# The Influence of Therapy Enriched with the Erigo^®^Pro Table and Motor Imagery on the Body Balance of Patients After Stroke—A Randomized Observational Study

**DOI:** 10.3390/brainsci15030275

**Published:** 2025-03-05

**Authors:** Anna Olczak, Raquel Carvalho, Adam Stępień, Józef Mróz

**Affiliations:** 1Faculty of Rehabilitation, Józef Piłsudski University of Physical Education in Warsaw, 00-968 Warsaw, Poland; 2RISE-Health—Escola Superior de Tecnologias da Saúde do Tâmega e Sousa (ESTeSTS) Instituto Politécnico de Saúde do Norte, Avenida Central de Gandra, 1317, 4585-116 Paredes, Portugal; raquel.carvalho@ipsn.cespu.pt; 3Neurological Clinic, Military Institute of Medicine-National Research Institute, 128 Szaserów Street, 04-141 Warsaw, Poland; astepien@wim.mil.pl; 4Rehabilitation Clinic, Military Institute of Medicine-National Research Institute, 128 Szaserów Street, 04-141 Warsaw, Poland; jmroz@wim.mil.pl

**Keywords:** stroke, motor imaging, Erigo^®^Pro tilt table, balance, physiotherapy

## Abstract

Purpose: Impaired balance leads to loss of function, e.g., the inability to walk safely. Therefore, restoring balance is a common goal of rehabilitation after a stroke. An innovative motor imaging and robotic device, the Erigo^®^Pro walking table, was used to improve balance in patients who had suffered an acute stroke. Materials and Methods: Sixty-six stroke patients in the acute phase with an average age of 64.85 ± 18.62 years were randomly assigned to one of three groups (22 subjects each) and treated with different therapies (conventional, conventional with Erigo^®^Pro, and conventional with Erigo^®^Pro enriched with motor imaging). The duration of therapy was two weeks. Patients were assessed before and after completion of therapy. The study used the trunk stability test and the Berg Balance Scale to assess balance, and the Riablo™ device to measure static balance. In addition, an assessment of the superficial tension of the transversus abdominis and multifidus muscles was performed. The clinical trial registration URL unique identifier was NCT06276075. Results: In each of the groups studied, the therapies applied resulted in significant improvement in functional assessment of trunk stability and balance (TCT < 0.001 and BBS < 0.001). The assessment of balance in the frontal (*p* = 0.023) and sagittal (*p* = 0.074) planes with the Riablo™ device confirmed the superiority of motor imaging-enhanced therapy at the level of a statistical trend. The tension of the transversus abdominis was higher at the second measurement (M = 14.41; SE = 3.31). Conclusions: Motor imagery-enhanced therapy is most important, both for trunk stability and functional improvement of body balance parameters and for increasing transversus abdominis muscle tension.

## 1. Introduction

Stroke is a common health problem, the second most common cause of death, and a leading cause of adult disability in Europe [1]. Recent advances have revealed that strokes that affect the cerebrum can cause more intricate disruptions in cognitive functions, including memory, decision-making, and postural and motor control. After a stroke, loss of balance is a common motor disorder that can make it difficult to perform everyday activities [2]. In addition, immobilization or sensory deprivation, which are common in neurological patients, can lead to dysfunction of the sensorimotor system due to the lack of proprioceptive input, causing changes at the cortical level and interfering with motor performance, for example during quiet standing [3,4].

Restoring balance is a common goal of stroke rehabilitation as it helps prevent falls and allows free movement [5,6]. Postural control is a complex process involving the central nervous system, the musculoskeletal system, and sensory information from the somatosensory, vestibular and visual systems [7].

In this study, we used motor imagery and the Erigo^®^Pro device in addition to conventional rehabilitation to assess the effects on patients’ balance.

New rehabilitation approaches such as motor imagery (MI) and/or robotic devices have already been used to improve balance in neurological patients [6,8,9].

MI requires the internal production of the visual and kinesthetic components of movement, which can be performed either from the first-person perspective (i.e., as if one were an actor in the action) or from the third-person perspective (i.e., as if one were a spectator of the motor task) but without execution of the movement [10,11].

The Erigo^®^Pro (Hocoma, Volketswil, Switzerland) is a tiltable robotic table with a walking function. It transfers the body weight to the lower limbs and allows verticalization up to 90°, combined with rhythmic passive or active alternating movements. This mechanism aims to improve proprioception and exteroceptive stimulation in the lower limbs in order to promote patient participation, such as cognition, and recovery from neurological disorders in patients with acquired brain injuries, such as improving balance and gait [12]. Many authors have compared the use of the Erigo^®^Pro table versus classical rehabilitation in patients after stroke, head injury or spinal cord injury [13,14,15,16]. The use of the Erigo^®^Pro table in patients with impaired consciousness has been studied [17,18], as well as the differences in the use of classical verticalization and verticalization with the Erigo^®^Pro [19]. In addition, research has shown that tactile sensations on the plantar part of the foot can have a positive influence on vertical orientation during standing [20,21,22]. This is a crucial factor in maintaining proper postural control. Although it is safe for stroke patients and improves muscle tone, the effects of this technique on balance are still controversial and further randomized clinical trials are needed [8].

The question therefore arises of whether sensory input from a robotic device (Erigo^®^Pro) and/or motor imagery can improve stability and balance in neurological patients who have experienced a stroke.

With this background, our study aims to compare the effects of conventional therapy, therapy with the Erigo^®^Pro tilt table, and motor imagery-enhanced therapy on the body balance of stroke patients.

## 2. Materials and Methods

### 2.1. Study Design

This was a randomized observational study. The study involved 66 stroke patients who were randomly assigned to one of 3 groups of 22 patients each. The group of eligible patients who received conventional physiotherapy was examined first. Allocation of participants to the other study groups (II, conventional physiotherapy and Erigo^®^Pro; and III, conventional physiotherapy, Erigo^®^Pro, and motor imaging), was based on a coin toss. The groups received different therapies, including conventional, conventional with the addition of the Erigo^®^Pro table, and an additional enrichment with motor imagery, and these were treated as the independent variables.

The patients were examined before and after completion of their therapy. Various tests were performed as part of the study, including the trunk stability test, the Berg Balance Scale, and a static balance assessment using the Riablo™ device, as well as the assessment of the superficial tension of the transversus abdominis and multifidus muscles. These were used as the dependent variables. The study was conducted according to protocol no. 9/KRN/2023, which is registered in the Clinical Trial Registration URL: http://www.clinicaltrials.gov (accessed on 18 February 2024). The unique identifier is NCT06276075.

### 2.2. Ethical Approval

The study was conducted in accordance with the recommendations of the Ethics Committee of the Military Medical Institute (MMI) in Warsaw, Poland, which approved the protocol, and written informed consent was obtained from all subjects in accordance with the principles of the Declaration of Helsinki (approval number 6/MMI/2020).

### 2.3. Participants

The study was conducted at the Department of Rehabilitation of the Military Medical Institute (MMI) in Warsaw, Poland. Post-stroke patients were recruited from the Department of Rehabilitation of the Military Medical Institute (MMI). Patients aged between 18 and 85 years were included in the study if they met the following inclusion criteria: a confirmed diagnosis of acute stroke, either ischemic or hemorrhagic, as determined through clinical evaluation or imaging, and hemodynamic stability. Furthermore, patients were required to possess sufficient cognitive function to comprehend and comply with simple instructions and engage proactively in the rehabilitation process. Patients also needed to tolerate positioning in either the supine or upright posture. Exclusion criteria were: persistent language and cognitive deficits (assessed by Montreal Cognitive Assessment (MoCA) and Mini Mental State Examination (MMSE)), inattention, visual impairment, depression, or inability to toilet themselves independently prior to their stroke. The presence of significant contraindications, including but not limited to fractures, spinal instability, or other orthopedic conditions, was also taken into consideration.

One hundred post-stroke patients were selected to participate in the study. Of these, 34 patients did not meet the inclusion criteria, including 29 due to their functional condition, and 5 people refused to participate in the study. The remaining 66 patients were recruited and divided into three groups, each of which received different treatments.

The full inclusion criteria were as follows: (1) patients between 30 and 85 years old, (2) a confirmed diagnosis of acute stroke, either ischemic or hemorrhagic, as determined through clinical evaluation or imaging, from 6 to 8 weeks after stroke, (3) hemodynamic stability, (4) patients with sufficient cognitive function, (5) no fractures, spinal instability or other orthopedic conditions, (6) patients who could stand and walk with or without assistance (modified Rankin scale = 3), with slight neurological deficits (NIHSS ≤ 7).

The full exclusion criteria were as follows: (1) less than six weeks after the stroke event, (2) epilepsy, (3) inability to sit and stand, (4) persistent deficits in speech and cognitive function, or lack of attention, (5) visual disturbances, (6) depression, (7) high or very low blood pressure, dizziness, or malaise.

The flow of the participants in the individual phases of the study is shown below (Figure 1).

Finally, 66 patients who had suffered a stroke (6–8 weeks post stroke) participated in the study. They were aged 38–85 years (mean 64.85 ± 18.62 years), and comprised 39 males and 27 females. The clinical evaluation of patients after a stroke was performed by the physician admitting the patient to the clinic on the day of admission. Biometric data are shown in Table 1.

### 2.4. Study Procedure

Sixty-six patients were randomly assigned to the following groups: conventional physiotherapy, conventional physiotherapy with Erigo^®^Pro, conventional physiotherapy with Erigo^®^Pro and motor imaging (MI). Patients who were admitted to the rehabilitation clinic, fulfilled the inclusion conditions and gave their written consent to undergo the study were consecutively included in the study groups.

In the conventional group (Group I) and the other groups, patients participated in daily physiotherapy, which included intensive preparatory training for standing and sitting positions, strengthening exercises, weight-bearing exercises, and balance and coordination exercises (totalling 40 to 60 min of therapy). The intervention was performed by a physiotherapist with more than 20 years of experience and who had undergone specialized training with the Erigo^®^Pro table. The procedure was carried out on an individual basis. The control group was included to ascertain whether the observed gains were attributable to the acute phase, i.e., spontaneous recovery, or whether the Erigo^®^Pro/motor imaging added any benefit to the rehabilitation process.

In the group with Erigo^®^Pro (Group II), the effective therapy included one session on a tilting table with an angle of 42° for 20 min, 5 times a week (Monday to Friday) at a speed of 32 steps per minute, followed by a physiotherapy session, for two weeks. The decision to allocate a fortnight for the program was based on the premise that this would provide an optimal temporal window for the induction of quantifiable enhancements in motor function, while concurrently ensuring the viability and logistical manageability of the intervention within the parameters of an acute care environment. The selected timeframe is pivotal in allowing for the repetition and intensity of both motor imagery and active physical therapy, which are pivotal in stimulating the neural circuits involved in motor recovery. Furthermore, the two-week duration corresponds with the clinical course of acute stroke patients, where intensive rehabilitation is commonly initiated soon after stabilization to maximize recovery potential. It also allows for close monitoring of patient responses and adaptation of therapy based on individual progress. Research also suggests that short-term, high-frequency rehabilitation during this early phase can yield significant benefits in terms of functional outcomes, particularly when combined with technologies such as the Riablo™ table, which provides biofeedback and real-time performance tracking. The decision to implement a two-week therapy duration was made to optimize patient engagement, facilitate early functional gains, and ensure the feasibility of the intervention within the limitations imposed by acute care while aligning with evidence-based practices in stroke rehabilitation.

In the motor imagery (MI) group (Group III), active imagery was introduced in addition to conventional physiotherapy during passive walking on the Erigo^®^Pro table. During the exercises on the table, the patients wore headphones that blocked out all external noise, closed their eyes, and had the task of imagining the sensation of their body moving (ego fantasy), differentiating the imagination of walking in different real environments (park, forest, beach, snow, city streets, walking at different speeds, climbing and descending stairs, running, etc.) [10,23].

Each patient was assessed before the intervention (pre-treatment (m1) and the day after the intervention (post-treatment (m2)—experimental condition), with a 2-week break between assessments. We also used BBS recordings for each patient, which allowed verification of the accuracy of the assessment by a blinded examiner (RC from Portugal).

### 2.5. Outcome Measures

Clinical assessment included the trunk control test (TCT), the Berg Balance Scale (BBS) [18,24,25], assessment of static balance in the sagittal and frontal planes on the Riablo™ device, and assessment of superficial tension of the multifidus and transversus abdominis (using the Luna EMG device, Gliwice, Poland) recorded during body balance tests on the Riablo™ device platform.

Erigo^®^Pro is a device used for early verticalization of the patient. It enables the stimulation of a physiological walking movement through a system of electronic actuators. The system for independent control of the verticalization and walking mechanisms enables precise adjustment of, among others, table setting parameters, such as tilt angle and speed of change of tilt angle. Erigo^®^Pro also allows the operator to set the parameters for the stimulation of the walking movement, i.e., movement pattern, amplitude, and speed of the walking movement (Figure 2) [13,14,19].

The Riablo™ device (CoRehab, Trento, Italy)—a multifunctional device for training and examining patients (with measurement error less than 5%)—was used to objectively measure static balance in the frontal and sagittal planes. A special pressure-sensitive platform, connected via wireless technology to a multimedia program that informs the patient in real time of his/her position relative to the center of the disk, measures the degree of deviation of the patient’s body from the balance position. Thanks to the biofeedback function, the patient can visually track the position of their body’s center of gravity on the laptop screen during the 2-min measurement and balance accordingly to be as close as possible to the center of the target (Figure 3) [26,27,28,29].

A Luna EMG (a rehabilitation diagnostic robot developed by EGZOTech, Gliwice, Poland) was used to measure muscle tension (measurement accuracy [−1–+1 V+/−1 µV]) [30,31].

Surface electrodes (disposable EMG electrodes measuring 55 and 40 mm; Sorimex, Toruń, Poland) were attached to the subject’s body using the surface myography for the non-invasive assessment of muscles (SENIAM) method on the transverse abdominal and multifidus muscles, on the side directly affected by the stroke. Before each exercise, the subject was instructed on how to perform the exercise.

### 2.6. Sample Size Calculation

The sample size was estimated a priori. The studies were planned based on the comparison of the results of three groups in two measurements and a moderate effect was assumed. The sample size was estimated using the program G * Power 3.1.9.4. With 3 groups and 2 measurements, the assumption of a moderate effect (f = 0.25), alpha = 0.05 and a test power of 0.95, the minimum sample size was 66 people (22 people per group).

### 2.7. Statistical Analysis

The analyses were performed in IBM SPSS Statistics 29.0 (Armonk, NY, USA: IBM Corp; 2022). In a first step, the basic measures of descriptive statistics were calculated and the Shapiro–Wilk test was used to check the conformity of the distribution of the results with the normal distribution. To compare the balance parameters, taking into account the type of therapy and measurement, an analysis of variance was performed in a mixed 3 × 2 design, where the between-subjects factor was the group membership (conventional group vs. ergogenic group vs. motor imagination) and the within-subjects factors were the two measurements of the balance parameters. The Bonferroni test was used as a post-hoc test. The significance level was α = 0.05. *p*-values ranging from 0.05 to 0.1 were treated as significant at the statistical trend level.

## 3. Results

Despite the deviation from the normal distribution of the analyzed variables, taking into account the equality of the compared groups and the fulfillment of the assumption of homogeneity of variances in the compared groups, further analysis was performed using a parametric test.

### 3.1. Clinical Results—Type of Therapy and Equilibrium Parameters

To determine whether the type of therapy influenced the results of the balance parameters, an analysis of variance was performed in a 3 × 2 mixed design, where the between-subjects factor was group membership (control group vs. Erigo^®^Pro group vs. motor imagination group) and the within-subjects factor was two measurements (the balance parameters). The models analyzed included two main factors, measurement and for group membership, as well as an interactive factor that took both factors into account simultaneously. A summary of the analyzed effects can be found in Table 2.

### 3.2. Instrumental Results—TCT

The analysis showed significant main effects for measurement and for group membership.

The post-hoc analysis showed that significantly higher results were achieved in the second measurement (M = 76.00; SE = 1.87) than in the first measurement (M = 53.68; SE = 2.19), irrespective of group membership. In the Erigo^®^Pro group (M = 56.14; SE = 3.36), the results for TCT were significantly lower than in the visualization group (M = 73.75; SE = 3.36; *p* = 0.001). There were no differences between the control group and Erigo^®^Pro or between the conventional and visualization groups (*p* > 0.05). The interaction effect was not significant, indicating that the simultaneous consideration of both factors did not cause any differences in the TCT values.

### 3.3. Instrumental Results—BBS

The analysis revealed a significant main effect for the time of measurement. In the first measurement, the BBS score (M = 45.68; SE = 1.29) was significantly lower than in the second measurement (M = 52.14; SE = 1.10).

The analysis of simple effects for the measurement showed significant differences between the measurements in each of the analyzed groups (in the control group: F(1,63) = 45.21; *p* < 0.001; η^2^ = 0.42; in the Erigo^®^Pro group: F(1,63) = 34.02; *p* < 0.001; η^2^ = 0.35; in the visualization group: F(1,63) = 22.08; *p* < 0.001; η^2^ = 0.26). In the first measurement, the BBS value for each group was significantly lower than in the second measurement.

### 3.4. Instrumental Results—Sagittal Static Balance [°]

The analysis showed a significant effect of the interaction between time of measurement and group membership at the level of a statistical trend. The analysis of simple effects for the time of measurement showed differences at the level of statistical tendency between the measurements in the visualization group, F(1,63) = 3.30; *p* = 0.074; η^2^ = 0.05. Higher results were obtained for the first measurement (M = 5.41; SE = 0.77) than for the second (M = 4.00; SE = 0.73). No differences were found between the measurements in the other two groups (in the control group: F(1,63) = 0.67; *p* = 0.415; η^2^ = 0.01; in the Erigo^®^Pro group: F(1,63) = 1.51; *p* = 0.223; η^2^ = 0.02). For the first measurement, F(2,63) = 2.10; *p* = 0.131; η^2^ = 0.06 and the second measurement, F(2,63) = 0.55; *p* = 0.581; η^2^ = 0.02, no differences were found between the groups.

### 3.5. Instrumental Results—Frontal Static Balance [°]

The analysis showed a significant effect of the interaction between time of measurement and group membership at the level of a statistical trend. The analysis of the simple effects for the groups showed no significant differences between the groups for either the first measurement, F(2,63) = 0.20; *p* = 0.821; η^2^ = 0.01 or the second measurement, F(2,63) = 1.66; *p* = 0.198; η^2^ = 0.05.

The analysis of the simple effects for the time of measurement revealed significant differences between the measurements in the visualization group, F(1,63) = 5.45; *p* = 0.023; η^2^ = 0.08. Higher results were obtained for the first measurement (M = 2.36; SE = 0.41) than for the second measurement (M = 1.50; SE = 0.36). No differences were found between the measurements in the other two groups (in the control group: F(1,63) = 0.38; *p* = 0.541; η^2^ = 0.01; in the Erigo^®^Pro group: F(1,63) = 0.97; *p* = 0.329; η^2^ = 0.01).

### 3.6. Instrumental Results—Voltage of the Transversus Abdominis Muscle [µV]

The analysis revealed a significant main effect for the time of measurement. In the first measurement (M = 8.12; SE = 0.95), the tension value was lower than in the second measurement (M = 12.26; SE = 1.91). The simple effects analysis for the group showed differences at the level of a statistical trend in the case of the first measurement, F(2,63) = 2.62; *p* = 0.081; η^2^ = 0.08. The tension level in the visualization group (M = 5.84; SE = 1.64) was lower than in the control group (M = 11.05; SE = 1.64; *p* = 0.086). No differences were found between the other groups. For the second measurement, there were also no differences between the groups, F(2,63) = 0.33; *p* = 0.721; η^2^ = 0.01. The analysis of the simple effects for the time of measurement showed a significant effect only in the visualization group, F(1,63) = 6.24; *p* = 0.015; η^2^ = 0.09. In the first measurement (M = 5.84; SE = 1.64), the tension was lower than in the second measurement (M = 14.41; SE = 3.31). In the control group, F(1,63) = 0.01; *p* = 0.912; η^2^ < 0.01 and in the Erigo^®^Pro group, F(1,63) = 1.75; *p* = 0.191; η^2^ = 0.03, these effects were insignificant.

For the tension of the multifidus muscle, none of the analyzed effects were statistically significant. This means that there were no significant differences in the parameter values between the measurements, between the groups, or when both factors were considered simultaneously.

## 4. Discussion

The aim of our study was to investigate the effect of motor imagery, tilt table intervention and conventional rehabilitation on the balance of post-stroke patients.

We analyzed trunk stability and body balance in all patients using the TCT, the BBS and body balance assessment procedures in the sagittal and frontal planes on the Riablo™ device platform. We extended the examination on the platform by examining the surface tension of the transversus abdominis and multifidi muscles to detect changes in the tension of the muscles that deeply stabilize the trunk. In both the TCT and BBS tests, we obtained significantly higher results in the second measurement in each of the groups tested. These tests are relatively commonly used by researchers to assess trunk stability and body balance [18,24,25]. In addition, post-hoc analysis of the TCT test revealed significantly higher scores in the motor imagery group compared to the Erigo^®^Pro group, with no significant differences between these groups and the control group. Similar research results in the area of BBS were obtained by researchers who performed trunk muscle stabilization exercises for 15 min daily in a separate experimental group [25], as well as in the area of TCT, where researchers assessed the trunk abilities of patients with non-acute and chronic stroke using the trunk control test, among others. In addition, measures of trunk stability were significantly related to the subjects’ balance, gait and functional performance [24]. The analysis of static balance performed with the Riablo™ again showed a significant effect of the interaction of measurement time and group membership in terms of statistical trend. In this case, significantly higher results were obtained in both the sagittal and frontal planes in the first measurement than in the second, indicating an improvement in balance in our patients. Higher results are correlated with greater body movements of the test subjects. The more stable the body, the better the static balance, i.e., the smaller the amplitude of the deflections in each of the tested planes.

The assessment of body balance (Equi-Test) carried out by Isabell V. Bonan et al. on a group of 40 patients at least 12 months after a stroke showed that the mean results for patients with hemiplegia were significantly lower than for healthy people. This led the researchers to conclude that many patients with hemiplegia are dependent on visual stimuli [32].

In our study, as the analysis of the simple measurement effects shows, we achieved statistically significant effects following therapy enriched with motor imagery.

Our results are similar to those reported the studies by Cho H. et al., who investigated the effect of motor imagination training on the balance and walking ability of stroke patients in the chronic phase. They studied a total of 28 subjects who were assigned to the following groups: an experimental group (n = 15—motor imagery training consisting of imagining normal walking movements for 15 min combined with gait training for 30 min, totaling 45 min per day, 3 times per week) and a control group (n = 13—gait training only for 30 min per day, 3 times per week). The balance assessment test, the functional reach test and the get-up-and-go test were performed over time. In these studies, all parameters increased significantly in the group in which motor imagery was used compared to the control group [33].

The last parameter analyzed, the assessment of the tension of the muscles that deeply stabilize the trunk, showed a significant main effect for the measurement only in the case of the transversus abdominis muscle, which obtained higher results in the second measurement in each of the groups studied. The analysis of the simple effects for the group again showed differences at the level of a statistical tendency. At the first measurement, the tension level was lower in the visualization group than in the control group, which seems to be important when evaluating therapies used in stroke patients, whose aim is to improve body balance. These results seem to be confirmed by a study on the importance of the tension of the muscles that deeply stabilize the trunk and the possibility of functional improvement in post-stroke patients without neurological disorders, in which a significantly higher tension of the multifidus and supraspinatus muscles was identified; however, the motor tasks evaluated were performed in sitting and standing positions and the importance of tension was analyzed. These muscles are used for the correct execution of movements in the frontal and sagittal planes and high knee lifts, when walking on the spot as fast as possible [31]. Similarly, the studies by Verheyden G. and the studies mentioned above investigated trunk function in patients with non-acute and chronic stroke in relation to balance, gait and functional ability. The results of these studies showed that measures of trunk rehabilitation were significantly related to the subjects’ balance, gait and functional performance scores [24].

Our results are also consistent with those of a previous randomized controlled trial that showed that trunk stability training (specifically progressive trunk stability training with selective pelvic exercises) had positive effects on trunk function, standing balance and mobility in a group of 32 subjects who had experienced a stroke [34]. Another randomized controlled trial with 80 stroke patients confirmed that trunk stabilization exercises in combination with conventional therapy improved trunk control, dynamic sitting balance, standing balance, and gait in stroke patients [35]. Lee et al. studied a group of people after a stroke to prove the effectiveness of stabilization exercises. They observed an effect of trunk stability training on changes in abdominal muscle thickness, balance, and gait, but no significant differences were found in gait scores and balance scale [36]. Furthermore, the strong association between balance measures and functional abilities emphasizes the importance of trunk rehabilitation, especially when a combination of MI and Erigo^®^Pro therapy is used [24].

It is worth noting that our observational study and concomitant therapy were conducted for only 2 weeks (i.e., 10 days). The decision to implement a 2-week therapy duration was made with the objective of optimizing patient engagement, facilitating early functional gains, and ensuring the feasibility of the intervention within the limitations imposed by acute care, while aligning with evidence-based practices in stroke rehabilitation.

However, most studies in which the authors describe the use of Erigo^®^Pro tablet for early verticalization (during the acute phase of the disease) are conducted for at least 30 days [13,14,19]. Accordingly, research by De Luca et al., conducted over approximately 8 weeks, provides empirical evidence supporting the hypothesis that exercise and motor stimulation, facilitated by the Erigo^®^Pro device, can modulate brain activity; additionally, the distinct alterations in alpha and beta bands signify the potential of Erigo^®^Pro as a valuable tool in promoting neural plasticity [37].

In addition, the studies mentioned above compare conventional therapy and therapy with the Erigo^®^Pro tablet. Our research also included a group in which we used motor imaging as a form of therapy. Similar to the study by Cho H. et al., we were able to show that motor imagery contributes to the improvement of motor skills, including body balance [33]. However, these researchers compared two groups, conventional and MI, and the patient groups were smaller than ours.

### Limitations

Statistical analysis revealed deviations from the normal distribution of several variables examined, but the analyses were performed using parametric tests. We decided that this was permissible because the comparison groups were of equal size (22 subjects each, which met the assumptions of the sample calculation) and the assumption of homogeneity of variances in the comparison groups was met.

In addition, the muscle tension on the directly affected side of the body was analyzed. It seems that the analysis of tension on both sides of the body could help to verify the hypothesis about the influence of differences in tension on both sides of the body, and the importance of central stabilization in improving stability and balance in post-stroke patients.

The comparison of the therapeutic effects in the three groups studied, taking into account the extent of the procedures applied in each group (group I—conventional physiotherapy, group II—extension with exercises on the Erigo^®^Pro table, group III—extension with motor imagery), is appropriate and allows correct conclusions to be drawn regarding the importance of the therapy applied. On the other hand, future research should use a different system to enrich the therapeutic regime with new therapies, in order to ultimately select the most effective one. Moreover, as Group II did not receive formal instruction in motor imagery, any unintentional imagery could not be systematically controlled or measured. However, the integration of video observation into motor imagery training has been demonstrated to enhance skill acquisition by refining mental representations, activating mirror neurons, improving error detection, and enhancing focus, motivation, and confidence. A limitation inherent in the study design is the variation in the total time spent on training across groups. While it is conceivable that the additional time spent with the Erigo^®^Pro Table and motor imagery may have introduced another variable, the study nevertheless provides valuable insight into whether these specific interventions offer benefits beyond those of conventional therapy.

## 5. Conclusions

In each of the groups studied, the therapies applied resulted in a significant improvement in the functional assessment of trunk stability and balance (according to the measures TCT and BBS).

The therapy enriched with motor imagery proved to be the most important for trunk stability and functional improvement in balance parameters, as well as for increasing transversus abdominis muscle tension, which correlates with the functional improvement achieved in post-stroke patients.

## Figures and Tables

**Figure 1 brainsci-15-00275-f001:**
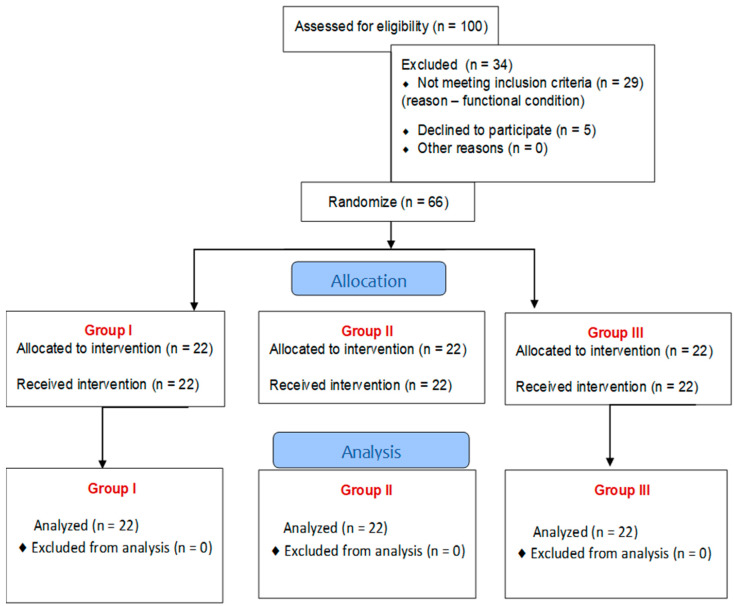
The flow of participants through each stage of the study.

**Figure 2 brainsci-15-00275-f002:**
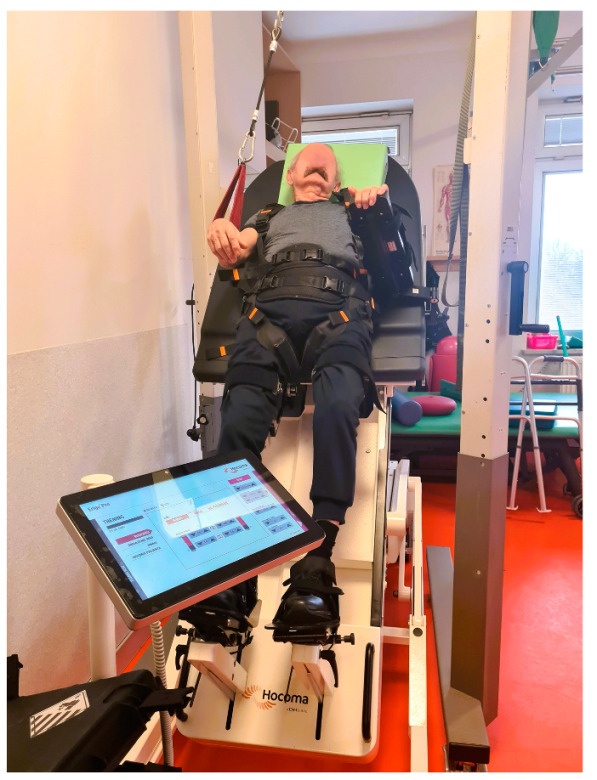
Erigo^®^Pro device in use during therapy.

**Figure 3 brainsci-15-00275-f003:**
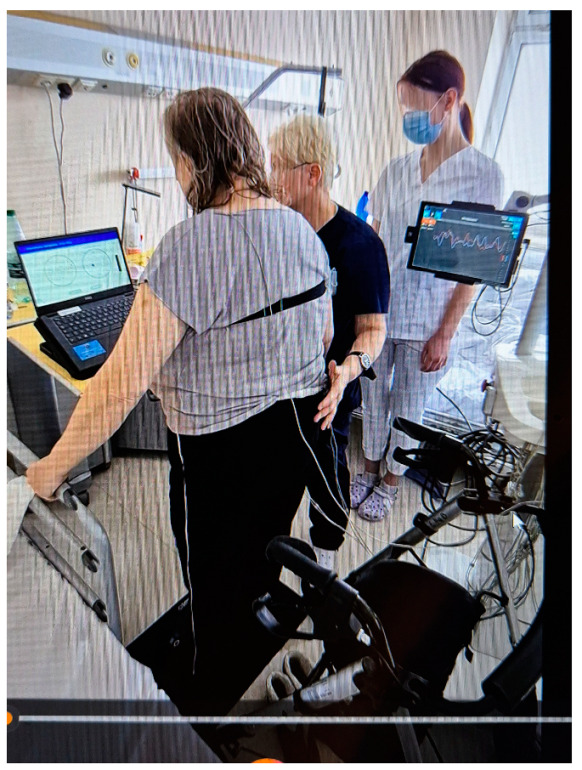
Riablo™ device with Luna EMG in use during examination of a patient.

**Table 1 brainsci-15-00275-t001:** Biometric data of study population and clinical control group.

Group		Age	Height	Body Mass	BMI
Conventional Therapy	Mean	66.73	167.00	74.27	26.60
	Sd.	7.65	9.16	9.73	2.61
Erigo^®^Pro	Mean	63.91	170.55	79.41	27.33
	Sd.	12.48	8.44	14.43	4.61
Erigo^®^Pro/MI	Mean	62.91	171.18	79.23	27.07
	Sd.	12.04	8.09	14.73	4.75
Kruskal–Wallis Test	H	1.08	1.95	2.03	0.29
	*p*	0.584	0.377	0.363	0.864
	Effect size	0.14	0.16	0.13	0.06

Legend: BMI (body mass index, calculated by dividing body weight in kilograms by the square of height in meters).

**Table 2 brainsci-15-00275-t002:** Summary of results of analysis of variance.

Parameter	Factors	*F*	*df*	*p*	η^2^
TCT	Measurement	323.11	1; 63	<0.001	0.84
	Group	6.87	1; 63	0.002	0.18
	Measurement * Group	0.52	2; 63	0.595	0.02
BBS [numerical score]	Measurement	99.25	1; 63	<0.001	0.61
	Group	1.67	1; 63	0.196	0.05
	Measurement * Group	1.03	2; 63	0.363	0.03
Static balance sagittal [°]	Measurement	0.02	1; 63	0.893	<0.01
	Group	0.86	1; 63	0.426	0.03
	Measurement * Group	2.74	2; 63	0.073	0.08
Static balance frontal [°]	Measurement	1.29	1; 63	0.260	0.02
	Group	0.27	1; 63	0.764	0.01
	Measurement * Group	2.75	2; 63	0.071	0.08
Voltage m multifidus [µV]	Measurement	0.44	1; 63	0.508	0.01
	Group	0.73	1; 63	0.487	0.02
	Measurement * Group	1.13	2; 63	0.330	0.03
Voltage m transverse [µV]	Measurement	4.59	1; 63	0.036	0.07
	Group	0.08	1; 63	0.921	0.01
	Measurement * Group	1.71	2; 63	0.190	0.05

BBS: Berg Balance Scale; TCT: trunk control test; * denotes the interaction between measurement and group membership.

## Data Availability

There are no restrictions on access to the data. The results of the study are in the possession of the corresponding author.

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
