# Peer review of "The Influence of Therapy Enriched with the Erigo®Pro Table and Motor Imagery on the Body Balance of Patients After Stroke—A Randomized Observational Study"

_brainsci, 2025, doi:10.3390/brainsci15030275_

Round 1
Reviewer 1 Report
Comments and Suggestions for Authors
Review report for “The influence of therapy enriched with the Erigo®Pro table 2 and motor imagery on the body balance of patients after stroke 3 - a randomized observational study.”
Comment to the authors;
Restoration of balance function is one of the essential goals of stroke rehabilitation. The authors investigated the effect of Ergo®Pro walking table and motor imagery with 66 post-stroke patients over 2 weeks. The authors applied well-established batteries for balance assessments to assess various aspects of balance. They found that the treatment regimen incorporating motor imagery induced a superior effect in terms of the static balance of the frontal and sagittal plane with the Riablo device assessment, while all the interventions induced functional recovery.
Whereas the most significant results show improvement over the time transition, which is just a natural course of the disease and treatment, the posthoc analysis of the TCT test revealed significantly higher results in the motor imagery+Erigo group compared to the Erigo-single treatment group. However, the present study included a critical problem in its design; while the conventional group received 40-60min training, the other groups received 60-80 min. Therefore, the specific effects of intervention groups cannot be extracted from this design. Another question is about motor imagery during Erigo training. Although the motor imagery group was facilitated to imagine things during training, persons in the other groups (Erigo single group) may imagine similar matters during their training. Generally, motor imagery studies prohibit actual movement and focus just on imagery; therefore, a fair comparison between the intervention and control groups is possible. In turn, this study utilized imagery together with actual movement. Therefore, there is the possibility that the participants in the control group imaged in a similar way to the intervention group unless the experimenters prohibited participants from imagining things, but it sounds impossible. Moreover, the allocation procedure of the conventional therapy group is different from that of the other groups.
Overall, the reviewer considers the current study to not satisfy the standard of a scientific article to be published.
Major concerns;
Abstract;
(lines 19-20) The grouping is unclear, with the current description being “conventional, with the addition of Erigo®Pro and enriched with motor imagery.”
The stroke phase (acute, subacute, chronic) to which the current study targeted is not described.
Introduction;
(lines 44-47) The involvement of Cerebrum is not described.
Materials and Methods
(lines 79-80) The rationale for investigating the control group first was not clear. Such an imbalanced intervention should involve a risk of selection and intervention bias. In addition, the method for choosing the conventional therapy group is not described.
(lines 80-84) Again, the description of the content of group III can cause misunderstanding among the readers: conventional + Erigo®pro table + motor imagery OR conventional + motor imagery. Please clarify it.
(lines 104-107) While the criteria include that acute phase patients, < 6w from onset, are not included in the trial, it did not distinguish subacute and chronic patients whose expectations for recovery differ a lot.
Table 1: The interval from the onset to intervention, modified Rankin scale, and NIHSS score at the onset must be included in this Table. In addition, the abbreviations should be spelled out.
(lines123-138) The study design is invalid in delineating the specific effect of ERIGO®Table and motor imagery because the total training time duration differs among groups; i.e., while the conventional group received 40- 60 min training, the other groups received 60-80 min.
(lines 132-138) While participants in Group III were facilitated to imagine during training, participants in Group II may imagine in a similar way. How did the authors prevent it?
Results section
Although the draft includes plenty of statistical values, the actual value, i.e., raw data for each assessment, is not presented.
The terminology “visualization group” requires explanation.
Minor concerns;
(line 180) Although people know IBM, the manufacturer information should be proposed as other devices.
Comments on the Quality of English Language
The reviewer felt that the quality of English should be improved.
Author Response
Manuscript ID: brainsci-3477172
Dear Reviewers,
Thank you very much for the analysis of our manuscript. We appreciate your comments and indication of fragments that should be corrected and explained. Considering your suggestions, all mistakes were corrected. The introduction of corrections and changes in the text caused the numbering of the lines to shift. In response to reviewers' comments, it provides the original numbering. To avoid misunderstandings, changes introduced in the text are marked in blue.
Reviewer #1:
Comments and Suggestions for Authors
Review report for “The influence of therapy enriched with the Erigo®Pro table 2 and motor imagery on the body balance of patients after stroke 3 - a randomized observational study.”
Comment to the authors;
Restoration of balance function is one of the essential goals of stroke rehabilitation. The authors investigated the effect of Ergo®Pro walking table and motor imagery with 66 post-stroke patients over 2 weeks. The authors applied well-established batteries for balance assessments to assess various aspects of balance. They found that the treatment regimen incorporating motor imagery induced a superior effect in terms of the static balance of the frontal and sagittal plane with the Riablo device assessment, while all the interventions induced functional recovery.
Whereas the most significant results show improvement over the time transition, which is just a natural course of the disease and treatment, the posthoc analysis of the TCT test revealed significantly higher results in the motor imagery+Erigo group compared to the Erigo-single treatment group. However, the present study included a critical problem in its design; while the conventional group received 40-60min training, the other groups received 60-80 min. Therefore, the specific effects of intervention groups cannot be extracted from this design. Another question is about motor imagery during Erigo training. Although the motor imagery group was facilitated to imagine things during training, persons in the other groups (Erigo single group) may imagine similar matters during their training. Generally, motor imagery studies prohibit actual movement and focus just on imagery; therefore, a fair comparison between the intervention and control groups is possible. In turn, this study utilized imagery together with actual movement. Therefore, there is the possibility that the participants in the control group imaged in a similar way to the intervention group unless the experimenters prohibited participants from imagining things, but it sounds impossible. Moreover, the allocation procedure of the conventional therapy group is different from that of the other groups.
Overall, the reviewer considers the current study to not satisfy the standard of a scientific article to be published.
Major concerns;
Abstract;
Comments 1: (lines 19-20) The grouping is unclear, with the current description being “conventional, with the addition of Erigo®Pro and enriched with motor imagery.”
Response 1: We would like to express our gratitude to the reviewer. It should be noted that all groups received conventional physiotherapy; however, one group was also administered the Erigo protocol, while another received the Erigo protocol in conjunction with motor imagery training. Due to the limitations imposed by the word count, we have endeavored to provide a more detailed explanation and have restructured the sentence to reflect the various therapies administered (conventional physiotherapy, conventional physiotherapy with Erigo®Pro, and both enriched with motor imagery).
Comments 2: The stroke phase (acute, subacute, chronic) to which the current study targeted is not described.
Response 2: We would like to express our appreciation for this suggestion and confirm that we have now included the patients in the acute phase in the manuscript, in the abstract, and together with other inclusion criteria.
Introduction;
Comments 3: (lines 44-47) The involvement of Cerebrum is not described.
Response 3: We would like to thank your suggestion. It is imperative to acknowledge that the study pertains to a cerebral occurrence. Consequently, we have incorporated the following addition: „Recent advances have revealed that strokes that affect the cerebrum can cause more intricate disruptions in cognitive functions, including memory, decision-making, and postural and motor control.”
Materials and Methods
Comments 4: (lines 79-80) The rationale for investigating the control group first was not clear. Such an imbalanced intervention should involve a risk of selection and intervention bias. In addition, the method for choosing the conventional therapy group is not described.
Response 4: All the groups participated in physiotherapy sessions, but in two groups the Erigo was utilized, and in one of these groups, motor imagination training was also incorporated during the training time on the Erigo. It is acknowledged that by incorporating the training, there was a slight increase in the total intervention time. However, the objective was to compare the two experimental groups, ensuring that both had the same overall intervention time. The control group was included to ascertain whether the observed gains were attributable to the acute phase, i.e. spontaneous recovery, or whether the Erigo and motor imagination added any benefit to the rehabilitation process.
We added the information to the manuscript.
Comments 5: (lines 80-84) Again, the description of the content of group III can cause misunderstanding among the readers: conventional + Erigo®pro table + motor imagery OR conventional + motor imagery. Please clarify it.
Response 5: The patients in the final group have undergone a range of therapeutic interventions, including conventional physiotherapy, Erigo, and MI. This information has been incorporated into the manuscript.
Comments 6: (lines 104-107) While the criteria include that acute phase patients, < 6w from onset, are not included in the trial, it did not distinguish subacute and chronic patients whose expectations for recovery differ a lot.
Response 6: The requisite information has now been incorporated into the manuscript. The facility under discussion is a unit that provides acute and sub-acute care, but patients in the acute phase were included in the study between 6 and 8 weeks after the stroke. Corrections have been made in the text, in each place where it is mentioned.
Thank you very much for drawing our attention to these inaccuracies.
Comments 7: Table 1: The interval from the onset to intervention, modified Rankin scale, and NIHSS score at the onset must be included in this Table. In addition, the abbreviations should be spelled out.
Response 7: Thank you for your comment. We took into account the modified Rankin scale and the NIHSS score and this information is included in the inclusion criteria. Unfortunately, at this time it is not possible to add data regarding the interval from symptom onset to intervention, the modified Rankin scale, and the NIHSS score at baseline. However, we will consider these parameters for future submissions or revisions. Regarding the abbreviations, I will correct them and ensure that they are all written in full.
Comments 8: (lines123-138) The study design is invalid in delineating the specific effect of ERIGO®Table and motor imagery because the total training time duration differs among groups; i.e., while the conventional group received 40- 60 min training, the other groups received 60-80 min.
Response 8: Whilst the observation that the total training time differed between groups is indeed correct, and constitutes a limitation in the study design, the inclusion of a control group served to ensure that any observed improvements were not merely a consequence of spontaneous recovery in the acute phase. This thus serves to isolate the effects of the interventions under study. Whilst the additional time spent with the ERIGO® Table and motor imagery may have introduced a variable, it nevertheless provides valuable insight into whether these specific interventions offer benefits beyond conventional therapy. It is acknowledged, however, that this discrepancy in time constitutes a limitation, and its consideration is imperative when interpreting the results.
We added to the limitation of the manuscript.
Comments 9: (lines 132-138) While participants in Group III were facilitated to imagine during training, participants in Group II may imagine in a similar way. How did the authors prevent it?
Response 9: We agree with the reviewer. While participants in Group III received explicit training on motor imagery and watched videos to guide their imagination during the ERIGO® sessions, it is possible that participants in Group II may have also engaged in spontaneous motor imagery during their training. However, since Group II did not receive formal instruction on motor imagery, any unintended imagination could not be systematically controlled or measured. This represents a limitation of the study. However, the integration of video observation into motor imagery training has been demonstrated to enhance skill acquisition by refining mental representations, activating mirror neurons, improving error detection, and enhancing focus, motivation, and confidence. This was added to the research limitations.
Results section
Comments 10: Although the draft includes plenty of statistical values, the actual value, i.e., raw data for each assessment, is not presented.
Response 10: Thank you for your question. The table with the actual values will be included in the appendix due to its length.
Table Supplement 1. Descriptive statistics for the analyzed balance parameters in the three groups.
M |
Me |
SD |
Sk. |
Kurt. |
Min. |
Maks. |
W |
p |
|
Control Group I |
|||||||||
Measurement 1 |
|
|
|
|
|
|
|
|
|
TCT [numerical score] |
52.91 |
48.00 |
13.98 |
0.94 |
0.33 |
36.00 |
87.00 |
0.85 |
0.003 |
BBS [numerical score] |
46.23 |
48.00 |
9.66 |
-0.93 |
0.50 |
22.00 |
58.00 |
0.92 |
0.086 |
Static balance sagittal [°] |
3.23 |
2.50 |
2.51 |
1.39 |
3.12 |
0.00 |
11.00 |
0.88 |
0.015 |
Static balance frontal [°] |
2.36 |
1.50 |
2.15 |
1.38 |
1.37 |
0.00 |
8.00 |
0.82 |
0.001 |
Voltage m multifidus [µV] |
19.42 |
16.96 |
13.39 |
1.43 |
2.40 |
5.36 |
59.11 |
0.87 |
0.009 |
Voltage m transverse [µV] |
11.05 |
7.00 |
11.24 |
2.72 |
8.41 |
2.36 |
52.18 |
0.67 |
<0.001 |
Measurement 2 |
|
|
|
|
|
|
|
|
|
TCT [numerical score ] |
76.36 |
74.00 |
12.45 |
0.32 |
-0.75 |
61.00 |
100.00 |
0.88 |
0.010 |
BBS [numerical score] |
53.77 |
52.50 |
7.71 |
0.83 |
0.70 |
43.00 |
74.00 |
0.94 |
0.199 |
Static balance sagittal [°] |
3.86 |
3.50 |
2.80 |
1.31 |
1.33 |
1.00 |
11.00 |
0.85 |
0.004 |
Static balance frontal [°] |
2.14 |
2.00 |
1.52 |
1.80 |
4.35 |
0.00 |
7.00 |
0.81 |
<0.001 |
Voltage m multifidus [µV] |
18.06 |
14.36 |
13.12 |
0.98 |
0.09 |
2.92 |
47.65 |
0.89 |
0.021 |
Voltage m transverse [µV] |
10.67 |
7.52 |
10.56 |
1.92 |
3.22 |
0.84 |
40.77 |
0.75 |
<0.001 |
Erigo Group II |
|||||||||
Measurement 1 |
|
|
|
|
|
|
|
|
|
TCT [numerical score] |
45.86 |
48.00 |
21.64 |
-0.26 |
-0.66 |
0.00 |
74.00 |
0.93 |
0.129 |
BBS [numerical score ] |
42.68 |
47.50 |
11.45 |
-1.41 |
1.66 |
11.00 |
57.00 |
0.86 |
0.005 |
Static balance sagittal [°] |
3.91 |
3.00 |
2.83 |
1.09 |
1.03 |
0.00 |
11.00 |
0.91 |
0.044 |
Static balance frontal [°] |
2.05 |
2.00 |
1.65 |
1.33 |
2.62 |
0.00 |
7.00 |
0.87 |
0.009 |
Voltage m multifidus [µV] |
15.20 |
14.60 |
5.64 |
-0.21 |
-0.95 |
4.63 |
23.71 |
0.96 |
0.533 |
Voltage m transverse [µV] |
7.47 |
5.74 |
6.66 |
3.06 |
11.53 |
0.32 |
33.48 |
0.68 |
<0.001 |
Measurement 2 |
|
|
|
|
|
|
|
|
|
TCT [numerical score] |
66.41 |
67.50 |
17.16 |
-0.28 |
-1.09 |
36.00 |
87.00 |
0.90 |
0.024 |
BBS [numerical score ] |
49.23 |
54.50 |
10.82 |
-1.49 |
1.51 |
21.00 |
59.00 |
0.80 |
<0.001 |
Static balance sagittal [°] |
4.86 |
3.00 |
4.27 |
1.65 |
2.39 |
1.00 |
17.00 |
0.80 |
<0.001 |
Static balance frontal [°] |
2.41 |
2.00 |
2.04 |
1.61 |
4.12 |
0.00 |
9.00 |
0.86 |
0.005 |
Voltage m multifidus [µV] |
15.86 |
14.09 |
9.56 |
1.46 |
2.72 |
4.28 |
43.71 |
0.87 |
0.008 |
Voltage m transverse [µV] |
12.01 |
6.13 |
15.62 |
3.11 |
11.02 |
2.43 |
72.45 |
0.60 |
<0.001 |
Visualization Group III |
|||||||||
Measurement 1 |
|
|
|
|
|
|
|
|
|
TCT [numerical score] |
62.27 |
61.00 |
16.83 |
0.28 |
-1.02 |
36.00 |
87.00 |
0.89 |
0.016 |
BBS [numerical score] |
48.14 |
52.50 |
10.26 |
-1.62 |
2.14 |
20.00 |
58.00 |
0.80 |
<0.001 |
Static balance sagittal [°] |
5.41 |
3.00 |
4.99 |
1.46 |
1.17 |
1.00 |
18.00 |
0.78 |
<0.001 |
Static balance frontal [°] |
2.36 |
2.00 |
1.99 |
1.80 |
5.07 |
0.00 |
9.00 |
0.84 |
0.002 |
Voltage m multifidus [µV] |
16.74 |
14.30 |
8.87 |
1.63 |
3.12 |
6.35 |
43.11 |
0.85 |
0.004 |
Voltage m transverse [µV] |
5.84 |
5.07 |
2.78 |
0.26 |
-0.70 |
1.06 |
10.75 |
0.95 |
0.322 |
Measurement 2 |
|
|
|
|
|
|
|
|
|
TCT [numerical score] |
85.23 |
87.00 |
15.69 |
-0.79 |
-0.22 |
48.00 |
100.00 |
0.85 |
0.003 |
BBS [numerical score] |
53.41 |
56.50 |
8.02 |
-1.79 |
2.90 |
30.00 |
60.00 |
0.77 |
<0.001 |
Static balance sagittal [°] |
4.00 |
3.50 |
3.06 |
0.83 |
-0.52 |
1.00 |
10.00 |
0.86 |
0.005 |
Static balance frontal [°] |
1.50 |
1.00 |
1.47 |
2.66 |
9.23 |
0.00 |
7.00 |
0.69 |
<0.001 |
Voltage m multifidus [µV] |
19.94 |
19.21 |
12.44 |
1.28 |
2.43 |
2.95 |
56.53 |
0.91 |
0.053 |
Voltage m transverse [µV] |
14.41 |
9.47 |
19.21 |
3.00 |
9.69 |
2.58 |
86.24 |
0.59 |
<0.001 |
Legend: normality test of distribution – Shapiro-Wilk
Comments 11: The terminology “visualization group” requires explanation.
Response 11: Since imagination was associated with observing the action in the video, we call them the visualization group, but thanks to your comment we realized that this explanation was flawed when we presented group III, and it has been corrected in the article.
Minor concerns;
Comments 12: (line 180) Although people know IBM, the manufacturer information should be proposed as other devices.
Response 12: Change has been made following your suggestion. It appears as a legend under table 1.
Dear reviewers, if we have missed anything or not done a thorough job of correcting something, please bring it to our attention.
Thank you very much for your time.

Reviewer 2 Report
Comments and Suggestions for Authors
This randomized observational study used the Erigo Pro walking table to enhance balance in stroke patients. Sixty-six stroke patients were allocated to three groups for a 2-week long therapy: conventional therapy, therapy supplemented with Erigo Pro, and therapy enhanced with motor imagery. Patients were evaluated before and after the therapy. The researchers discovered that the interventions made a big difference in the functional evaluation of trunk stability and balance in both of the groups they tested. Also, using the Riablo device to test balance in both the frontal and sagittal planes showed a statistically significant favorability for therapy combined with motor imagery.
The manuscript presents an important study, and the results would be of interest to the journal readers. The manuscript is generally well-written, but the authors should strongly consider the following suggestions/comments to improve its overall quality.
1. What was the inclusion criteria used for this study? Include the details of the inclusion criteria in the manuscript.
2. What was the rationale for choosing a two-week therapy duration? Include this information in the manuscript to benefit the readers.
3. Lines 210 - 211: "Between the control group and Ergio and there were no differences between the conventional and visualization groups (p > 0.05)." This statement is not clear.
4. Line 301: "The last analyzed one". What do the authors mean by this phrase?
5. Were there any incidents during the study? If so, include the details in the manuscript.
6. Did any of the users from the intervention groups give feedback on the walking table device? Was the feedback in line with the results of the study?
7. Including a few relevant images (e.g., Erigo Pro being used, display from Riablo device, etc.) from the study might benefit the reader.
Comments on the Quality of English LanguageThe manuscript needs to be revised for English grammar and style.
Author Response
Manuscript ID: brainsci-3477172
Dear Reviewers,
Thank you very much for the analysis of our manuscript. We appreciate your comments and indication of fragments that should be corrected and explained. Considering your suggestions, all mistakes were corrected. The introduction of corrections and changes in the text caused the numbering of the lines to shift. In response to reviewers' comments, it provides the original numbering. To avoid misunderstandings, changes introduced in the text are marked in blue.
Reviewer #2:
This randomized observational study used the Erigo Pro walking table to enhance balance in stroke patients. Sixty-six stroke patients were allocated to three groups for a 2-week long therapy: conventional therapy, therapy supplemented with Erigo Pro, and therapy enhanced with motor imagery. Patients were evaluated before and after the therapy. The researchers discovered that the interventions made a big difference in the functional evaluation of trunk stability and balance in both of the groups they tested. Also, using the Riablo device to test balance in both the frontal and sagittal planes showed a statistically significant favorability for therapy combined with motor imagery.
The manuscript presents an important study, and the results would be of interest to the journal readers. The manuscript is generally well-written, but the authors should strongly consider the following suggestions/comments to improve its overall quality.
Response: Thank you very much for the thorough analysis of our manuscript.
The following comments and answers:
Comments 1: What was the inclusion criteria used for this study? Include the details of the inclusion criteria in the manuscript.
Response 1: I did not clearly define the inclusion criteria, which, in their current form, appear before the exclusion criteria.
Inclusion criteria: 1) patients between 30 and 85 years old, 2) a confirmed diagnosis of acute stroke, either ischemic or hemorrhagic, as determined through clinical evalua-tion or imaging, from 6 to 8 weeks after stroke, 3) hemodynamic stability, 4) patients with sufficient cognitive function, 5) no fractures, spinal instability or other orthope-dic conditions, 6) patients who can stand and walk with or without assistance (modi-fied Rankin scale = 3), with slight neurological deficits (NIHSS ≤7).
Comments 2: What was the rationale for choosing a two-week therapy duration? Include this information in the manuscript to benefit the readers.
Response 2: The decision to allocate a fortnight for the program was based on the premise that this would provide an optimal temporal window for the induction of quantifiable enhancements in motor function, while concurrently ensuring the viability and logistical manageability of the intervention within the parameters of an acute care environment. The selected timeframe is pivotal in allowing for the repetition and intensity of both motor imagery and active physical therapy, which are pivotal in stimulating the neural circuits implicated in motor recovery.
Furthermore, the two-week duration corresponds with the clinical course of acute stroke patients, where intensive rehabilitation is commonly initiated soon after stabilization to maximize recovery potential. It also allows for close monitoring of patient responses and adaptation of therapy based on individual progress. Research also suggests that short-term, high-frequency rehabilitation during this early phase can yield significant benefits in terms of functional outcomes, particularly when combined with technologies such as the Riablo Table, which provides biofeedback and real-time performance tracking. The decision to implement a two-week therapy duration was made to optimize patient engagement, facilitate early functional gains, and ensure the feasibility of the intervention within the limitations imposed by acute care while aligning with evidence-based practices in stroke rehabilitation.
We have added the explanation presented above in the Study procedure section. Thank you for drawing our attention to this inaccuracy.
Comments 3: Lines 210 - 211: "Between the control group and Ergio and there were no differences between the conventional and visualization groups (p > 0.05)." This statement is not clear.
Response 3: Thank you very much for this attention. This incorrect word order has been corrected. Currently, after corrections, it is line 262-263 as follows: "There were no differences between the control group and Erigo and between the conventional and visualization groups (p > 0.05)."
Comments 4: Line 301: "The last analyzed one". What do the authors mean by this phrase?
Response 4: We should clarify the term. It should read "The last parameter analyzed". We have corrected, in line 357. Thank you.
Comments 5: Were there any incidents during the study? If so, include the details in the manuscript.
Response 5: Throughout the designated study period, no incidents were observed to have transpired.
Comments 6: Did any of the users from the intervention groups give feedback on the walking table device? Was the feedback in line with the results of the study?
Response 6: Throughout the study, participants expressed appreciation for the protocol; however, no formal opinion was documented.
Comments 7: Including a few relevant images (e.g., Erigo Pro being used, display from Riablo device, etc.) from the study might benefit the reader.
Response 7: We have attached a photo of Erigo during use and an image from the Riablo device during patient balance assessment. Section Outcome measures.
Thank you for bringing this to our attention.
Comments 8: Comments on the Quality of English Language
The manuscript needs to be revised for English grammar and style.
Response 8: The manuscript has been thoroughly revised for English grammar and style by a native speaker. Revisions have been made throughout the entire document to ensure clarity, coherence, and correct language usage
Dear reviewers, if we have missed anything or not done a thorough job of correcting something, please bring it to our attention.
Thank you very much for your time.

Reviewer 3 Report
Comments and Suggestions for Authors
General comments
I think that this is a very interesting topic. However, my biggest concern about this study is that the authors did not provide any clinical details about the post-stroke patients. I would like to know whether the enrolled patients had ischemic or hemorrhagic strokes, how long ago they experienced the stroke, and the severity of their condition. For example, were the patients able to walk independently? If so, rehabilitation with the Erigo would seem unnecessary, as this tool is typically used in the acute phase of stroke or head injury to prevent hypotensive episodes and maintain gait pattern activation in the brain during gradual verticalization.
Specific comments:
Line 37-38. Revise carefully the definition of stroke and its impact on healthcare/society costs and patient's quality of life.
Study design paragraph. Here you should report where patients were enrolled, how many patients, and basic information about your sample (gender and mean age). In addition, you should describe carefully the study design, which is not clear. Please revise it carefully. To help you, you can use CONSORT checklist for randomised studies.
You should add a paragraph about the randomization procedures.
lines 85-86. Outcome measures need to be reported in the specific section and not here.
Please state clearly the inclusion and the exclusion criteria of the participants.
Table 1 is not clear at all. The first column should report the demographic and clinical features, the second column should report the data of all participants, third, fourth and fifth columns should report descriptive statistics (ie. mean/sd) of each demographical and clinical feature of each group, the sixth column should report the p-value related to the comparison among the three groups. Under the table put a legend reporting the type of the statistical test.
Line 141. What is BBS? If it is referred to the clinical scale, you have to report below in the appropriate section.
Lines. 146-147. Why did you register the EMG of only two muscles? and why these two?
Add a figure of Riablo and Luna EMG set-up.
How do you analyse the EMG signals? Specify it in the statistical analysis section.
Table 2. Instead of "points" I would use "numerical score" for both BBS and TCT.
I suggest to revise the results section carefully and dividing it into two main subsections, like "clinical results" and "instrumental results", as an example.
line 257. "For multifidus muscle tension," why it is in bold style?
line 264. use the acronyms for the clinical scales to be consistent.
I suggest implementing the discussion section with these two refs that showed the role of Erigo in improving plasticity in patients with acquired brain injury: 10.3390/brainsci14040319, 10.3390/biomedicines1210224. These two works could aid you in strengthening your claims in the discussion.
line 344. Instead of this title, use a simpler way: "4.1 Limitations"
Author Response
Manuscript ID: brainsci-3477172
Dear Reviewers,
Thank you very much for the analysis of our manuscript. We appreciate your comments and indication of fragments that should be corrected and explained. Considering your suggestions, all mistakes were corrected. The introduction of corrections and changes in the text caused the numbering of the lines to shift. In response to reviewers' comments, it provides the original numbering. To avoid misunderstandings, changes introduced in the text are marked in blue.
Reviewer #3:
General comments
I think that this is a very interesting topic. However, my biggest concern about this study is that the authors did not provide any clinical details about the post-stroke patients. I would like to know whether the enrolled patients had ischemic or hemorrhagic strokes, how long ago they experienced the stroke, and the severity of their condition. For example, were the patients able to walk independently? If so, rehabilitation with the Erigo would seem unnecessary, as this tool is typically used in the acute phase of stroke or head injury to prevent hypotensive episodes and maintain gait pattern activation in the brain during gradual verticalization.
Response: The decision to implement a two-week therapy duration was made to optimize patient engagement, facilitate early functional gains, and ensure the feasibility of the intervention within the limitations imposed by acute care while aligning with evidence-based practices in stroke rehabilitation. Research also suggests that short-term, high-frequency rehabilitation during this early phase can yield significant benefits in terms of functional outcomes. Thank you for this comment.
Specific comments:
Comments 1: Line 37-38. Revise carefully the definition of stroke and its impact on healthcare/society costs and patient's quality of life.
Response 1: I changed the suggested sentence slightly. Thank you very much for your comment.
Comments 2: Study design paragraph. Here you should report where patients were enrolled, how many patients, and basic information about your sample (gender and mean age). In addition, you should describe carefully the study design, which is not clear. Please revise it carefully. To help you, you can use CONSORT checklist for randomised studies.
You should add a paragraph about the randomization procedures.
Response 2: The study design paragraph has been revised to take into account the randomization of study subjects (line: 87, "we used allocation based on a coin toss"). In turn, detailed information about the recruitment of study participants and themselves is provided in subsection 2.3. Participants.
Thank you very much for all your suggestions.
Comments 3: lines 85-86. Outcome measures need to be reported in the specific section and not here.
Response 3: I'd like to thank the reviewer for his/her suggestion a specific section has been created for this purpose, line 182.
Comments 4: Please state clearly the inclusion and the exclusion criteria of the participants.
Response 4: I did not clearly define the inclusion criteria, which, in their current form, appear before the exclusion criteria.
Inclusion criteria: 1) patients between 30 and 85 years old, 2) a confirmed diagnosis of acute stroke, either ischemic or hemorrhagic, as determined through clinical evalua-tion or imaging, from 6 to 8 weeks after stroke, 3) hemodynamic stability, 4) patients with sufficient cognitive function, 5) no fractures, spinal instability or other orthope-dic conditions, 6) patients who can stand and walk with or without assistance (modi-fied Rankin scale = 3), with slight neurological deficits (NIHSS ≤7).
Comments 5: Table 1 is not clear at all. The first column should report the demographic and clinical features, the second column should report the data of all participants, third, fourth and fifth columns should report descriptive statistics (ie. mean/sd) of each demographical and clinical feature of each group, the sixth column should report the p-value related to the comparison among the three groups. Under the table put a legend reporting the type of the statistical test.
Response 5: In Table 1 we have presented an analysis of basic patient data such as age, body mass, and Body Mass Index, additionally, at the request of another reviewer, I have added an explanation of BMI in the legend to the table. The table proposed by you could be created, but should we duplicate the information in the text, especially since we explain that the study included only people in the acute phase of stroke plus additional explanations about the patient's clinical condition? If necessary, we will create a table according to your suggestions. Thank you very much for this comment.
Comments 6: Line 141. What is BBS? If it is referred to the clinical scale, you have to report below in the appropriate section.
Response 6: I'd like to thank the reviewer for his/her suggestion it was added to a specific section. For an explanation, see Outcome measures, line: 183.
Comments 7: Lines. 146-147. Why did you register the EMG of only two muscles? and why these two?
Response 7: The selection of these two muscles was made on the basis of their critical function in the maintenance of postural control and trunk stability. It has been demonstrated that weakness in these muscles can have a detrimental effect on balance. In the Outcome measures section, we have added a photo showing the patient's examination (Riablo - recording body balance and Luna EMG - tension of selected muscles).
Comments 8: How do you analyse the EMG signals? Specify it in the statistical analysis section.
Response 8: We used the Luna EMG device to read and analyse the EMG signal. The Luna EMG device provides a record of, among other things, the average values ​​of muscle tension in microvolts [µV]. We used this record to analyse the data of the subjects studied. The explanation of the electrode arrangement can be found in subsection 2.5. Outcome measures.
Thank you very much for this comment.
Comments 9: Table 2. Instead of "points" I would use "numerical score" for both BBS and TCT.
Response 9: We would like to express our gratitude for your contribution in the form of a suggestion, which has now been received.
Comments 10: I suggest to revise the results section carefully and dividing it into two main subsections, like "clinical results" and "instrumental results", as an example.
Response 10: Thank you very much, we followed the advice. However, we left the section layout as before but added the proposed definitions of the presented results.
We hope that it is equally legible.
Comments 11: line 257. "For multifidus muscle tension," why it is in bold style?
Response 11: The issue was a formatting error, which has since been rectified.
Comments 12: line 264. use the acronyms for the clinical scales to be consistent.
Response 12: I agree, thank you very much for this comment, the changes have been made, line 319.
Comments 13: I suggest implementing the discussion section with these two refs that showed the role of Erigo in improving plasticity in patients with acquired brain injury: 10.3390/brainsci14040319, 10.3390/biomedicines1210224. These two works could aid you in strengthening your claims in the discussion.
Response 13: Thank you very much for this comment and tip. We used one suggestion that enriched our discussion, as I hope (line 395 -399). Unfortunately, we were unable to find the second one. Perhaps the record is not complete?
Thank you for your help.
Comments 14: line 344. Instead of this title, use a simpler way: "4.1 Limitations"
Response 14: We would like to say thank you for your suggestion, which we have now received.
Dear reviewers, if we have missed anything or not done a thorough job of correcting something, please bring it to our attention.
Thank you very much for your time and very thorough analysis of our manuscript.

Round 2
Reviewer 1 Report
Comments and Suggestions for Authors
The reviewer considers that the issues were appropriately amended. This article is ready for publication.